# Spatio-Temporal Analysis of Simulated Summer Extreme Precipitation Events under RCP4.5 Scenario in the Middle and Lower Reaches of the Yangtze River Basin

**Lu Liu** [1]📷, **Weiyi Sun** [1] **and Jian Liu** [1,2,3,*]

[1] Key Laboratory for Virtual Geographic Environment, Ministry of Education, State Key Laboratory Cultivation Base of Geographical Environment Evolution of Jiangsu Province, Jiangsu Center for Collaborative Innovation in Geographical Information Resource Development and Application, School of Geography Science, Nanjing Normal University, Nanjing 210023, China; 191301014@njnu.edu.cn (L.L.); weiyisun@njnu.edu.cn (W.S.)

[2] Jiangsu Provincial Key Laboratory for Numerical Simulation of Large Scale Complex Systems, School of Mathematical Science, Nanjing Normal University, Nanjing 210023, China

[3] Open Studio for the Simulation of Ocean-Climate-Isotope, Qingdao National Laboratory for Marine Science and Technology, Qingdao 266237, China

* Correspondence: jliu@njnu.edu.cn

**Abstract:** In the context of global warming, the frequency and intensity of extreme climate events, especially extreme precipitation events, have increased. The middle and lower reaches of the Yangtze River Basin are important areas for economic development, and are also one of the areas where rainstorms and flood disasters frequently occur in China. Improving the prediction of future summer extreme precipitation in this region under the greenhouse gas emission pathway that aligns with sustainable economic development (Representative Concentration Pathway 4.5, RCP4.5) will help decision-makers better cope with the impact of increased natural disasters, such as floods. The medium-resolution CESM1.0 (Community Earth System Model 1.0) data ($1° \times 1°$) has limitations in capturing regional climate differences. Therefore, we designed a downscale experiment using the WRF3.8 (Weather Research and Forecasting 3.8) model to obtain the daily summer precipitation grid data with $0.25° \times 0.25°$ latitude and longitude resolution over the middle and lower reaches of the Yangtze River Basin from May to September in 2006–2030 (WRF025). The research shows that the WRF025 data is reliable in simulating the summer extreme precipitation events over the middle and lower reaches of the Yangtze River Basin, especially in the lower reaches of the Yangtze River. Compared to CESM1.0 data, WRF025 data significantly improves the ability to simulate the numerical value and distribution of summer extreme precipitation in the middle and lower reaches of the Yangtze River. Under the RCP4.5 scenario, compared to 2006–2014, there is no significant change in daily summer precipitation in the middle and lower reaches of the Yangtze River Basin during 2015–2030, with a significant decrease in daily summer extreme precipitation. There are significant regional differences in spatial distribution, with a significant decrease in Hunan and Hubei, and a significant increase in Jiangxi and Fujian. Under high-pressure control, the lower reaches of the Yangtze River are dominated by downdraft, resulting in more sunny days and less precipitation. The increase (decrease) in water vapor transport and divergence may be the reason for the increase (decrease) in extreme precipitation. The most direct factor leading to an increase (decrease) in extreme precipitation is the vertical movement upwards (downwards). Furthermore, the anomalous descent (ascent) can be well explained by the easterly (westerly) wind anomaly on the southern (northern) side of the anomalous anticyclone via the isentropic gliding mechanism.

**Keywords:** summer extreme precipitation; WRF; RCP4.5

## 1. Introduction

In recent years, research on extreme weather and extreme climate events has received more and more attention due to their growing influence on the economy, the environment

and society [1,2]. Extreme precipitation events are an important part of extreme climate events, which often cause floods, mudslides and landslides. Increasingly more studies have shown that increased extreme precipitation events in many regions of the world will change the hydrological cycle and evapotranspiration, which have serious social impacts [3–8]. Against the background of global warming, changes in extreme climate events, especially future extreme climate events, have attracted both widespread domestic and international attention. Predicting the temporal and spatial variation characteristics of future extreme precipitation events can help decision-makers better cope with the impact of increasing extreme precipitation events.

Using climate system models that set different emission scenarios to predict climate and extreme climate change has become a research hotspot [9,10]. The Representative Concentration Pathway (RCP) future scenarios (RCPs; typical concentration paths) characterized by stable concentrations were adopted by the Fifth Assessment Report (AR5) of the Intergovernmental Panel on Climate Change (IPCC) [11,12]. RCP4.5 is an intermediate stable path, and its time trend is more consistent with future economic development, particularly in China. In addition, the new generation Global Climate Model (GCMs) in the Coupled Model Intercomparison Project Phase 5 (CMIP5) can provide valid data for the AR5 of IPCC. However, the GCMs can only give the monthly (seasonal, annual) average state on a global scale. For extreme climate events, especially extreme precipitation events, high-resolution models are required for simulation [13,14]. Therefore, downscaling using various techniques is necessary. Dynamic downscaling is one of the most important downscaling methods, with several advantages: (1) clear physical meaning; (2) can be used anywhere without being affected by observational data; (3) can also be used for different resolutions. For example, a multi-RCM ensemble project has been designed by the National Institute of Meteorological Research (NIMR) under the Korea Meteorological Administration (KMA) that participated in the CMIP5 with the Hadley Global Environment Model 2-Atmosphere (HadGEM2-AO) simulations through RCP scenarios [15]. Based on the daily dataset output of this multi-RCM, Joong-Bae Ahn et al. (2016) found that there was a significant increase in heavy rains between 2046–2070 and 2076–2100 compared to 1898–2005 over the Korean Peninsula [16]. Gao et al. (2014) analyzed the extreme climate events over the Tarim River Basin using the high-resolution data output by the Cosmo Model in Climate Mode (CCLM) driven by the Max Planck Institute Earth System Model LR (MPI-ESM-LR) from the German Max Planck Institute for Meteorology [17]. They found that the strong precipitation in 2016–2035 will be reduced by 7.88 mm compared to 1986–2005. Based on the daily precipitation data (0.5 × 0.5) output from the Regional Climate Model system (RegCM4.0) driven by the Beijing Climate Center Climate System Model (BCC_CSM1.1), Wu and Huang (2016) found that the maximum expected increase in precipitation extreme intensity over the Pearl River Basin during the two future periods (2021–2050 and 2051–2080) will mainly occur in the southern region, compared to 1971–2000 [18]. Gao et al. (2014) found that the extreme precipitation centers in the Huai River Basin do not change, using the regional climate model CCLM driven by ECHAM6 (an atmospheric component of the MPI-M Earth System Model: ECHAM6) [17]. The center is still distributed in the south bank and coastal areas of the basin, while the precipitation intensity in the north shore and the inland areas of the basin is small.

The middle and lower reaches of the Yangtze River Basin are not only the main agricultural bases of China, but also the regions with strong social and economic development in China. At the same time, the middle and lower reaches of the Yangtze River Basin are also the regions with the most severe rainfall and flood disasters in China. For example, rainstorms in the Yangtze River valley during the summer of 1998 caused 4150 human deaths, 6.85 million collapsed houses and 166 billion RMB in economic losses. Therefore, it is valuable to study the seasonality of summer extreme precipitation in the middle and lower reaches of the Yangtze River Basin. The analysis in this paper will help predict future extreme precipitation, and assist decision-makers in better coping with the impact of increasing summer extreme precipitation events.

There have been many studies on the dynamic downscaling of the middle and lower reaches of the Yangtze River. Using the Weather Research and Forecasting (WRF) model, Xie et al. (2018) analyzed the correlation between a strong precipitation and the low-pressure intensity of the plateau in the summer of 2006 [19]; Liang et al. (2018) analyzed the two rainstorm processes in the middle and lower reaches of the Yangtze River during the Meiyu period [20]; the organization characteristics and triggering conditions of the mesoscale convective systems of the Meiyu front rainstorm in the middle and lower reaches of the Yangtze River were analyzed [21]; Chen et al. (2016) analyzed the characteristics of rainstorms in warm regions of the middle and lower reaches of the Yangtze River [22].

However, what are: (1) the summer extreme precipitation events over the middle and lower reaches of the Yangtze River Basin in 2006–2030 under the Rcp4.5 scenario based on WRF? (2) the characteristics of time and space changes in 2015–2030? (3) the physical mechanism of climate change? At the same time, for policymakers, the extreme climate change of the next decade is highly realistic. There has been less research on the above issues, so this research has certain scientific and practical significance.

## 2. Data and Methods

### 2.1. Configuration of the WRF Model

We selected the experimental data from 2006 to 2030 under the RCP4.5 scenario of the CMIP5 Earth System Model 1 output provided by the NCAR website to provide initial and side boundary conditions for the WRF. This dataset includes global bias-corrected climate model output data from version 1 of NCAR's Community Earth System Model (CESM1.0). The dataset contains all of the variables needed for the initial and boundary conditions for simulations with the WRF model or the Model for Prediction Across Scales (MPAS), provided in the Intermediate File Format specific to WRF and MPAS. While an ensemble of CESM simulations was performed for each scenario to characterize model uncertainty, only output from ensemble member 6—also known as the "Mother of All Runs (MOAR)"—was used to construct the present files, because that is the only member that has the full three-dimensional fields required to force WRF or MPAS available at six hourly intervals. The data were interpolated to 26 pressure levels and were provided in files at six hourly intervals. The variables were bias-corrected using the European Centre for Medium-Range Weather Forecasts (ECMWF) Interim Reanalysis (ERA-Interim) fields for 1981–2005, following the method by Bruyere et al. (2014) [23]. Files were available for a 20th Century simulation (1951–2005) and three concomitant RCP future scenarios (RCP4.5, RCP6.0 and RCP8.5) spanning 2006–2100. Its spatial resolution was $1° \times 1°$ and the time resolution was 6 h. The parameterization scheme selection is shown in Table 1. The advantage of the WSM3 simple ice scheme is that it greatly reduces computational complexity. Under limited computational conditions, it is possible to use the simple ice scheme for high-resolution numerical simulations.

Based on the first spatial model of the total summer extreme precipitation REOF in China from 1961 to 2014 (Figure 1), we found that the middle and lower reaches of the Yangtze River were positively centered: 27° N–32° N, 108° E–122° E. Considering the discontinuity of the boundary, when designing the downscaling experiment, we appropriately expanded the downscaling area to 26° N–32.5° N, 105° E–123° E. In this paper, the numerical simulation was conducted using WRF3.8, with the ARW framework selected, and a two-layer grid nested. The first layer covered most parts of eastern China, with 29.5° N and 115° E as the center, containing 46 grid points in the east-west direction and 43 grid points in the north-south direction, with a horizontal resolution of the area at 75 km. The second layer had 82 grid points in the east-west direction and 31 grid points in the north-south direction, with a horizontal resolution of the area of 25 km. The vertical layering was 30 layers with an integration step size of 120 s (Figure 2). The simulation time was from 00 UTC on 1 May to 00 UTC on 1 September (MJJAS) from 2006 to 2030 (WRF025).

**Table 1.** Selection of parameterization scheme.

| Physical Plan | Plan |
| --- | --- |
| Microphysical process | WSM3 simple ice [24] |
| Long wave radiation | Rrtm [25] |
| Short wave radiation | Dudhia [26] |
| Near ground level | Monin-Obukhov [27] |
| Land surface process | Noah land surface proce [28] |
| Boundary layer | YSU [29] |
| Cumulus parameterization | Shallow convection Kain-Fritsch [30] |

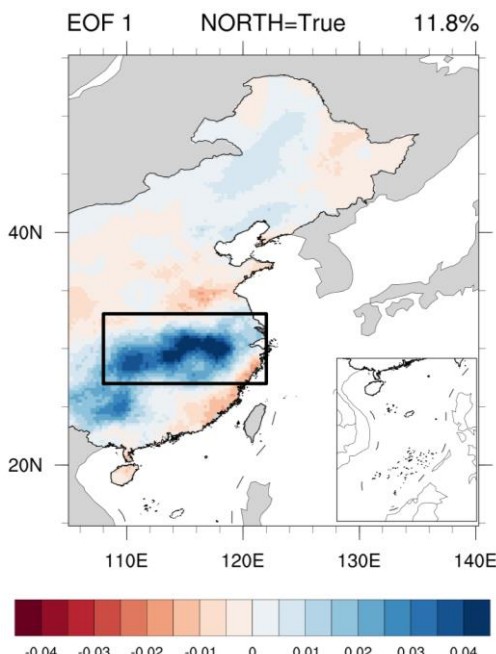

**Figure 1.** Spatial modes of the first REOF based on the analysis of 54-year (1961–2014) summer extreme precipitation in China (contours).

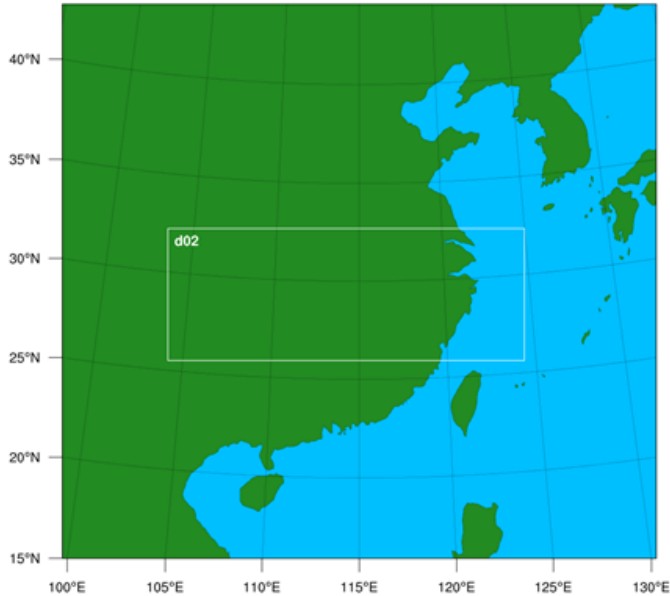

**Figure 2.** Simulation area and grid nesting. d02: domain 2. White box: 26° N–32.5° N, 105° E–123° E.

*2.2. Data*

The daily precipitation observation data came from the CN05.1 dataset provided by the China National Meteorological Administration (CNCC, also known as BCC). The spatial resolution is 0.25° × 0.25°, the time range is from 1961 to 2014 and the spatial range is the whole China region: 14.75° N–55.25° N, 69.75° E–140.25° E. The CN05.1 dataset was constructed by interpolating site data from 2416 observatories in China and was quality controlled [31].

The Tropical Rainfall Measuring Mission (TRMM) data product 3B42 has a spatial range of 50° S to 50° N, 180° W to 180° E, a spatial resolution of 0.25° × 0.25° and a time resolution of 3 h from 31 December 1997 to 1 January 2020. The data includes precipitation intensity and error. The 3B42 incorporates multiple microwave remote sensing data, including a TRMM Microwave Imager (TMI) on the TRMM satellite and a Special Sensor Microwave/Image (SSM/I) on the Defense Meteorological Satellite Program (DMSP). Because the data product incorporates multiple microwave remote sensing data and the data quality is higher than previous data products, it has become a data product recommended for scientific research [32]. The daily-scale precipitation calculates the daily precipitation under Beijing time through the daily precipitation intensity of eight times, and the monthly and annual precipitation is calculated by adding the daily precipitation.

The dataset including global bias-corrected climate model output data CN05.1 and TRMM data were used to compare with the WRF data to verify the simulation ability of the WRF025 data for the total summer extreme precipitation in the middle and lower reaches of the Yangtze River Basin. The set member 6 of CESM 1.0 was used as pre-scaling data to compare with the WRF025 data (after scaling) to explore whether the WRF025 data had any advantages (Table 2).

**Table 2.** Data.

| No. | Name | Spatial Resolution | Time Resolution | Period |
|-----|------|--------------------|-----------------|--------|
| 1 | CN05.1 | 0.25° × 0.25° | daily | 1961–2014 |
| 2 | TRMM | 0.25° × 0.25° | 3-h | 1997–2020 |
| 3 | WRF025 | 0.25° × 0.25° | 6-h | 2006–2030 |
| 4 | CESM1.0 (ensemble 6) | 1° × 1° | 6-h | 2006–2030 |

*2.3. Definition of Summer Extreme Precipitation*

To consider geographical differences, we used the 95th percentile method to define extreme precipitation thresholds [33,34]. Summer extreme precipitation events are defined as daily precipitation events with daily precipitation greater than the 95th percentile of summer precipitation. This definition has been widely used in recent research [2,33,35]. In this study, the 95th percentile threshold was calculated using the daily rainfall record for the entire summer (JJA) from 1961 to 2014.

The steps are as follows:

Step 1: Extract daily precipitation data with daily precipitation of greater than 0.0 mm in the middle and lower reaches of the Yangtze River from 1961 to 2014;

Step 2: Sort the data extracted from Step 1 from the minimum value of precipitation to the maximum value of precipitation;

Step 3: Extract the precipitation threshold corresponding to the 95th percentile of the sorted data;

Step 4: Define the daily precipitation events over the middle and lower reaches of the Yangtze River in the summer of 1961–2014 as the summer extreme precipitation events over the middle and lower reaches of the Yangtze River.

## 3. Results

### 3.1. Definition of the Middle and Lower Reaches of the Yangtze River Basin

In this paper, we used the rotational empirical orthogonal decomposition function (REOF) to analyze the spatial pattern of extreme summer precipitation (Figure 1). The spatial distribution structure separated by the traditional empirical orthogonal decomposition function (EOF) cannot clearly represent the characteristics of different geographical regions. In addition, in the empirical orthogonal decomposition expansion, the spatial distribution pattern of the feature vector will be different due to the difference in the range of the obtained regions, which causes difficulties in physical interpretation. These limitations can be overcome when using the REOF. The typical spatial distribution of REOF is clearer, which can reflect the climate changes of different regions. The sampling error of REOF is also much smaller than that of EOF. Therefore, the REOF has been given increasingly more attention.

Figure 1 shows the first spatial mode of the REOF of the total summer extreme precipitation in China from 1961 to 2014. The interpretation variance of this mode is 11.8%, and the mode passes the north test (true). From the first spatial mode of the REOF, it can be seen that it exhibits a negative-positive-negative three-pole distribution state in eastern China, in which the middle and lower reaches of the Yangtze River show a distinct positive center, while the North China region and the coastal areas of South China are negative centers. This mode is consistent with a summer extreme precipitation pattern obtained by another researcher [36]. Therefore, the range of the middle and lower reaches of the Yangtze River selected in this paper is the area with a positive center: 27° N–32° N, 108° E–122° E.

### 3.2. Verification of WRF025

We evaluated the WRF025 data by comparing and analyzing the spatial distribution mode, spatial correlation coefficient and standard deviation ratio (standard deviation of the model data/standard deviation of the observed data) of the summer precipitation and the summer extreme precipitation in the middle and lower reaches of the Yangtze River from 2006 to 2014. The same analysis was applied to the TRMM data, CESM1.0 (ensemble 6) data.

Figures 3 and 4 show the spatial distribution characteristics of summer precipitation amount and summer extreme precipitation amount over the middle and lower reaches of the Yangtze River Basin from 2006 to 2014 with CN05.1 data, TRMM data, CESM1.0 (ensemble 6) data and WRF025 data. Whether it is CN05.1 data, TRMM data or WRF025 data, CESM1.0 (ensemble 6) data, the values of summer precipitation amount and summer extreme precipitation amount differ, but the spatial distributions and high value centers of the summer precipitation amount and the summer extreme precipitation amount are very similar. As shown in Figure 4, the WRF025 data can generally simulate the spatial distribution characteristics of the summer extreme precipitation amount in the middle and lower reaches of the Yangtze River Basin. The high value of the summer extreme precipitation amount at the junction of Anhui, Jiangxi and Zhejiang, the junction of Hubei and Hunan, the junction of Zhejiang and Fujian and the junction of Hunan and Hubei can be well simulated. At the same time, compared to the CESM1.0 (ensemble 6) data, the WRF025 data can better simulate the higher summer extreme precipitation values in the southeastern part of Hubei and central Zhejiang, similar to the CN05.1data and TRMM data. However, the high value of the summer extreme precipitation amount over the middle and lower reaches of the Yangtze River Basin in the CN05.1 data shows a north-south distribution, while the WR025 data shows an east-west distribution; the highest value in WRF025 of the summer extreme precipitation amount in the middle and lower reaches of the Yangtze River is located in western Zhejiang, while the highest value in CN05.1 is located at the junction of Fujian and Zhejiang and the highest value in TRMM is located at the junction of Fujian and Jiangxi. The spatial distribution of CESM1.0 (ensemble 6) data greatly differs from CN05.1 and TRMM data, with high-value centers distributed in a zonal pattern from southwest to northeast. In summary, we believe that compared to CESM1.0

(ensemble 6) data, whether in the high value or high-value distribution range, WRF025 data can better capture the characteristics of the lower Yangtze River region consistent with CN05.1 and TRMM data.

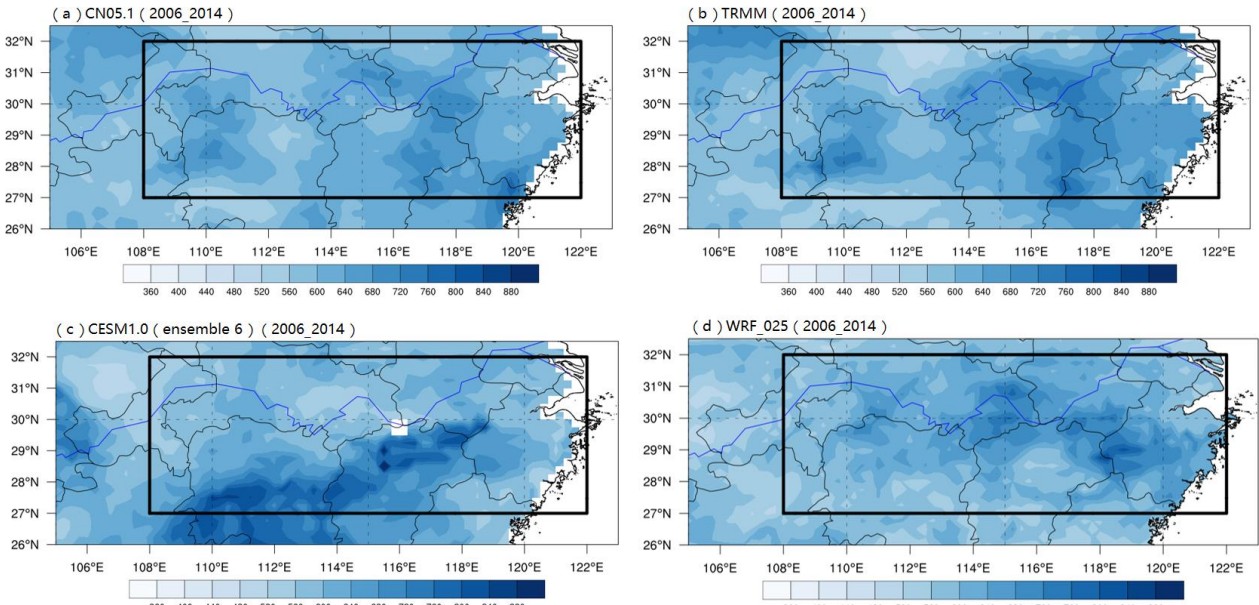

**Figure 3.** Spatial distribution characteristics of summer precipitation amount in the middle and lower reaches of the Yangtze River Basin during 2006–2014 from (**a**) CN05.1 data, (**b**) TRMM data, (**c**) CESM1.0 (ensemble 6) and (**d**) WRF025 data (unit: mm).

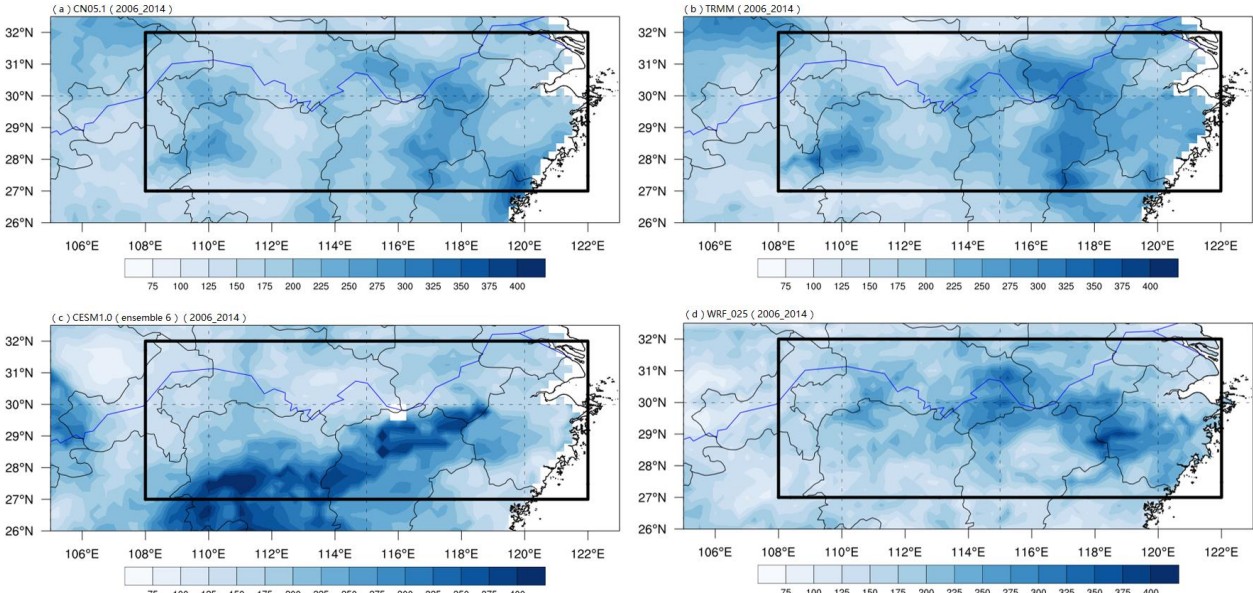

**Figure 4.** Spatial distribution characteristics of summer extreme precipitation amount in the middle and lower reaches of the Yangtze River Basin during 2006–2014 from (**a**) CN05.1 data, (**b**) TRMM data, (**c**) CESM1.0 (ensemble 6) and (**d**) WRF025 data (unit: mm).

At the same time, we calculated the spatial correlation coefficient and the standard deviation ratio of the WRF025 data and CESM1.0 (ensemble 6) compared to the CN05.1 and TRMM data for the summer precipitation amount and the summer extreme precipitation amount in the middle and lower reaches of the Yangtze River from 2006 to 2014 (Table 3). Compared with CESM1.0 (ensemble 6) data, the spatial correlation coefficients of WRF025 data with CN05.1 and TRMM data are both higher, and the ratio of standard deviation is

closer to 1. For example, for the summer extreme precipitation of WRF025 data, the spatial correlation coefficient is 0.97 ($p < 0.01$), and the ratio of standard deviation is 1.01, which is very close to the standard deviation of the CN05.1 data. At the same time, the spatial correlation coefficient is 0.96 ($p < 0.01$), and the ratio of standard deviation is 0.95, which is very close to the standard deviation of the TRMM data. For CESM1.0 (ensemble 6), the spatial correlation coefficients are 0.23 and 0.31, and the standard deviation ratios were 1.54 and 1.36, which are significantly different from the CN05.1 data and TRMM data. Therefore, we think that the simulation skill of WRF025 is reliable in the summer precipitation amount and the summer extreme precipitation amount over the middle and lower reaches of the Yangtze River.

**Table 3.** The spatial correlation coefficient and the standard deviation ratio.

| Variable | Data | CN05.1 (The Spatial Correlation Coefficient) | TRMM (The Spatial Correlation Coefficient) | CN05.1 (The Standard Deviation Ratio) | TRMM (The Standard Deviation Ratio) |
|---|---|---|---|---|---|
| **Summer precipitation** | CESM1.0 (ensemble 6) | 0.21 ($p > 0.1$) | 0.30 ($p > 0.1$) | 1.58 | 1.42 |
| | WRF025 | 0.98 ($p < 0.01$) | 0.97 ($p < 0.01$) | 1.02 | 0.97 |
| **Summer extreme precipitation** | CESM1.0 (ensemble 6) | 0.23 ($p > 0.1$) | 0.31 ($p > 0.1$) | 1.54 | 1.36 |
| | WRF025 | 0.97 ($p < 0.01$) | 0.96 ($p < 0.01$) | 1.01 | 0.95 |

*3.3. Climate Change in 2015–2030 Compared to 2006–2014*

Since the deadline for the CN05.1 data we used was 2014, we took 2014 as the cut-off year to study climate change and its circulation field characteristics between 2015–2030 compared to 2006–2014. Based on the total summer precipitation in the middle and lower reaches of the Yangtze River in WRF025 from 1961 to 2014, the change in total summer extreme precipitation (Figure 5) in the middle and lower reaches of the WRF025 Yangtze River from 2015 to 2030 was analyzed.

The difference between summer extreme precipitation over the middle and lower reaches of the Yangtze River Basin in WRF025 from 2015 to 2030 and from 1961 to 2014 (Figure 5b) shows that the main areas of decreased summer extreme precipitation are located in the north of Hunan, southeastern Hubei and Anhui, while the main areas of increased summer extreme precipitation are located in central Hunan, Jiangxi and Fujian.

In addition, we calculated the probability distributions of summer daily precipitation and extreme precipitation over the middle and lower reaches of the Yangtze River Basin in 2006–2014 and 2015–2030 (Figure 6). It can be seen that, compared with 2006–2014, the daily precipitation value of the middle and lower reaches of the Yangtze River in 2015–2030 has no obvious movement ($p > 0.10$), but the daily extreme precipitation value obviously shifts to the left, indicating that the extreme precipitation value from 2015 to 2030 is significantly smaller ($p < 0.05$).

Firstly, we examined the changes in the 500 hPa geopotential height field (2015–2030 minus 2006–2014), which were tested for significance ($p < 0.05$). The differential distribution of the 500 hPa geopotential height field indicates that compared to 2006–2014, the 500 hPa geopotential height field in the summer of 2015–2030 was significantly enhanced in the Hubei and Hunan regions, while it was significantly weakened in the Jiangxi and Fujian regions. Under the control of high pressure, the lower reaches of the Yangtze River basin are mainly dominated by downdraft, with more sunny weather and less precipitation, which is not conducive to the increase in extreme precipitation.

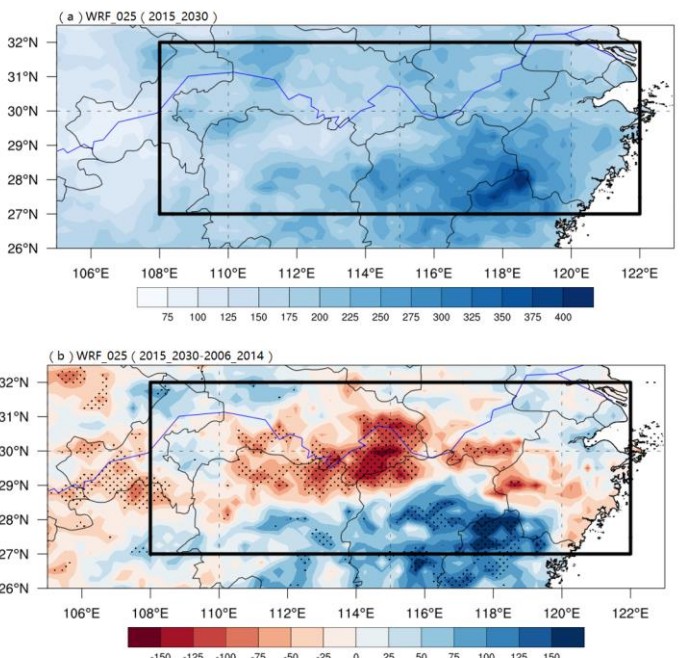

**Figure 5.** Spatial distribution of summer extreme precipitation amount over the middle and lower reaches of the Yangtze River Basin in WRF025 from (**a**) 2015 to 2030, and (**b**) the differences between 2015–2030 and 2006–2014 (unit: mm). Dots indicate the anomalies significant at the 0.05 confidence level.

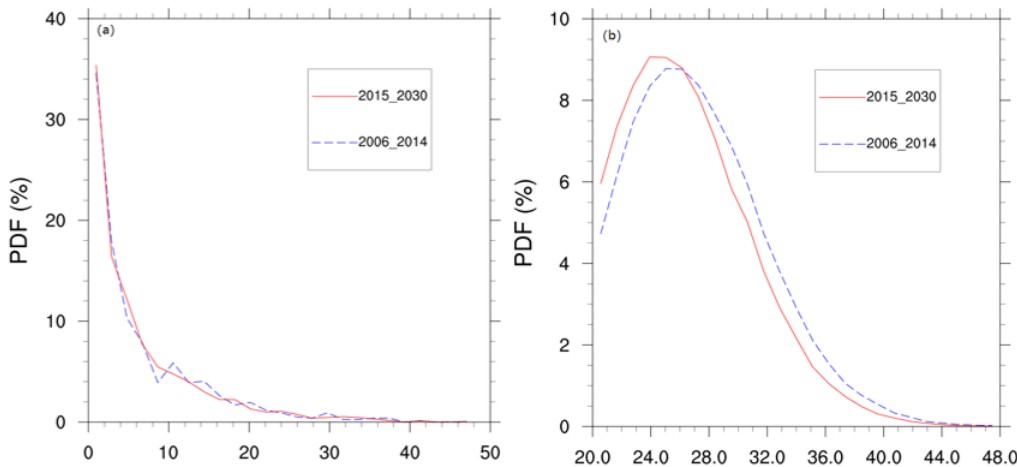

**Figure 6.** The probability distributions of (**a**) daily summer precipitation and (**b**) daily summer extreme precipitation in WRF025 in 2006–2014 (blue dashed line) and 2015–2030 (red solid line).

This article analyzes water vapor budget characteristics that are related to precipitation: water vapor transport (Figure 7d) and divergence (Figure 7e). It can be clearly observed that compared to 2006–2014, the water vapor transport to Hubei and Hunan regions significantly decreased during the 2015–2030 period, while it significantly increased in the Jiangxi and Fujian regions. At the same time, with the convergence of high-level water vapor in the Hubei and Hunan regions and the divergence of water vapor in the Jiangxi and Fujian regions, these water vapor budget processes may lead to a decrease in summer extreme precipitation in the Hubei and Hunan regions, with an increase in summer extreme precipitation in the Jiangxi and Fujian regions. However, based on the definition of water vapor transport, water vapor transport is stronger if the wind is stronger as specific humidity varies little, but it does not necessarily convert into precipitation. A more accurate diagnostic framework for precipitation variability is water vapor budget

analysis [37,38]. Many previous works have simplified the water vapor budget equation and have shown that vertical velocity at mid-troposphere dominates precipitation variability and the uncertainty of future precipitation change [39,40], rather than the water vapor amount.

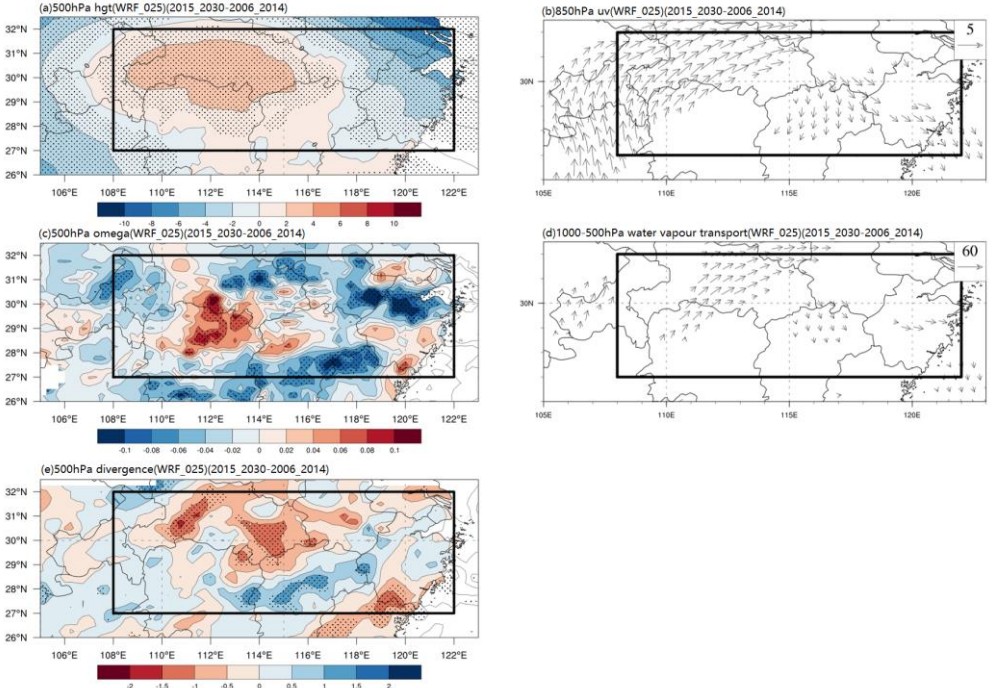

**Figure 7.** The difference distribution of (**a**) the 500 hPa geopotential height field (unit: m), (**b**) the 850 hPa wind field (unit: m s$^{-1}$), (**c**) the 500 hPa vertical velocity field (unit: Pa s$^{-1}$), (**d**) the 1000–500 hPa water vapor transport field (unit: kg m$^{-1}$ s$^{-1}$) and (**e**) the 500 hPa divergence field (unit: kg m$^{-2}$ s$^{-1}$) between 2015–2030 and 2006–2014 in summer extreme precipitation event days (WRF025). Dots indicate the anomalies significant at the 0.05 confidence level.

Therefore, the 850 hPa wind field (Figure 7b) and the 500 hPa vertical velocity field (Figure 7c) (2015–2030 minus 2006–2014) were obtained, which were tested for significance ($p < 0.05$). The difference distribution of the 500 hPa vertical velocity field shows that there is a significant positive center over Hubei and Hunan in the summer of 2015–2030, indicating that there is significant subsidence movement, which is not conducive to the formation of extreme precipitation. Furthermore, there is a significant negative center over Jiangxi and Fujian in the summer of 2015–2030, indicating that there is a significant upward movement, which is conducive to the formation of extreme precipitation. At the same time, the differential distribution of the 850 hPa wind field indicates that, compared to 2006–2014, the 850 hPa wind field in the summer of 2015–2030 has significant anticyclones in the Hubei and Hunan regions, while there are cyclones in the Jiangxi and Fujian regions. In the free troposphere, vertical motion (and its anomaly) is mostly generated by quasi-horizontal motion moving along sloping isentropic surfaces [41–43]. In winter, isentropic surfaces incline northward with altitude over the YRB, and a southerly (northerly) wind anomaly generates an ascent (descent) anomaly, just as in most mid-latitude regions [44]. However, in summer, there is a climatological warm center over the Tibetan Plateau and the isentropic surfaces incline eastward with altitude over the YRB; therefore, a westerly (easterly) wind anomaly stimulates an ascent (descent) anomaly over the YRB region in summer [44]. As seen in Figure 7b,c, the anomalous descent (ascent) is located on the southern (northern) flank of the anomalous anticyclone where anomalous easterly (westerly) wind prevails, so the anomalous descent (ascent) in Figure 7c can be well explained by the easterly (westerly) wind anomaly on the southern (northern) side of the anomalous anticyclone via the isentropic gliding mechanism [44].

## 4. Discussion

Due to the limitations of the research area, we are unable to explore the atmosphere-ocean interaction using WRF025 data. As an explanation, several studies have suggested that the Pacific Decadal Oscillation (PDO) may have had an impact on summer precipitation in East Asia after the 1970s. For example, Yang and Lau (2004) found a relationship between positive PDO and the decreasing trend in summer precipitation in North China [45]. In addition, Zhu et al. (2011) found that after 2000, summer precipitation in eastern China changed, with a decrease in precipitation in the Yangtze River Delta region and an increase in precipitation in central North China. Zhu et al. (2011) attributed these precipitation changes to the westerly changes brought about by PDO [46]. Other studies have also emphasized the potential regulation of PDO on EASM and related summer precipitation changes. For example, Zhou et al. (2013) showed that there has been an inverse correlation between EASM and PDO since 1950, where the negative correlation of PDO corresponds to stronger EASM and more precipitation in North China [47]. If our calculation conditions can be optimized, we may be able to use high-resolution numerical simulation results to further explore the impact mechanism of PDO on summer extreme precipitation in the middle and lower reaches of the Yangtze River basin.

The spatio-temporal variation characteristics of summer extreme precipitation events over the middle and lower reaches of the Yangtze River Basin from 2015 to 2030 are mainly based on the WRF_025 results. However, WRF_025 is only the result of one case. Therefore, uncertainty may occur. Gao et al. (2011) pointed out that multi-model and multi-set simulation is an important method for reducing uncertainty [48]. In the future, it is necessary to conduct multi-model and multi-set simulations on the basis of improved models to obtain more reliable information on future extreme precipitation changes in the middle and lower reaches of the Yangtze River basin in China.

Furthermore, one of the important methods to improve resolution is to assimilate conventional observation data with high-resolution remote sensing observation data, such as weather radars and meteorological satellites. Compared with simple downscaling, the research shows that the assimilation data of radar satellite high-resolution data can significantly improve the short-term prediction within 6 h of the model [49,50]. Therefore, while improving the model resolution, we should also pay more attention to the assimilation of high-resolution data.

## 5. Conclusions

This paper mainly analyzes the temporal and spatial variation characteristics and climate change physical mechanisms of summer extreme precipitation events over the middle and lower reaches of the Yangtze River Basin from 2015 to 2030 under the RCP4.5 scenario. The results show that:

(1) We validated the simulation capability of WRF025 for the summer extreme precipitation over the middle and lower reaches of the Yangtze River Basin during 2006–2014 using observations. The spatial correlation coefficients were 0.97 and 0.96 ($p < 0.01$). The standard deviation ratios were 1.01 and 0.95, so the WRF025 data was reliable. In terms of numerical value, the WRF025 data were closer to the CN05.1 data and the TRMM data. In distribution, WRF025 data could better capture the high-value distribution area in the lower reaches of the Yangtze River.

(2) Compared to 2006–2014, there was no significant change in daily precipitation in the middle and lower reaches of the Yangtze River during 2015–2030, but the daily extreme precipitation significantly moved to a smaller value. There were significant regional differences in the spatial distribution of extreme summer precipitation, which were related to the climate characteristics under the control of high pressure (low pressure). At the same time, water vapor transport/divergence also increased/diverged in areas where extreme precipitation increased in summer, and decreased/converged in areas where extreme precipitation decreased in summer. Furthermore, the anomalous descent (ascent) could be

well explained by the easterly (westerly) wind anomaly on the southern (northern) side of the anomalous anticyclone via the isentropic gliding mechanism.

**Author Contributions:** Conceptualization, L.L., W.S. and J.L.; methodology, L.L., W.S. and J.L.; formal analysis, L.L.; data curation, L.L.; writing—original draft preparation, L.L.; writing—review and editing, W.S. and J.L.; visualization, L.L.; supervision, W.S. and J.L.; funding acquisition, J.L. All authors have read and agreed to the published version of the manuscript.

**Funding:** This research was funded by the National Natural Science Foundation of China (Grant Nos. 42130604, 42105044, 41971108 and 42111530182), and the Priority Academic Development Program of Jiangsu Higher Education Institutions (Grant No. 164320H116).

**Data Availability Statement:** The data presented in this study are available on request from the corresponding author.

**Acknowledgments:** We thank Jia Wu (wujia@cma.gov.cn) from the National Climate Center, China Meteorological Administration, kindly provided the high-resolution observation data (CN05.1 data). More details about this data can be found at Wu and Gao [2013].

**Conflicts of Interest:** The authors declare no conflict of interest.

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
