# Peer review of "Spatio-Temporal Analysis of Simulated Summer Extreme Precipitation Events under RCP4.5 Scenario in the Middle and Lower Reaches of the Yangtze River Basin"

_sustainability, doi:10.3390/su15129218_

Round 1
Reviewer 1 Report
I have completed the review of your manuscript titled "Spatio-Temporal Analysis of Simulated Summer Extreme Precipitation Events under RCP4.5 Scenario in the Middle and Lower Reaches of the Yangtze River Basin." To provide detailed feedback and suggestions for improvement, I have prepared a document containing comments and recommendations.

NA
Reviewer 2 Report
Summary
This work performed a dynamic downscaling analysis of projected precipitation change in future over the Yangtze River Basin (YRB) region under RCP4.5 scenario. The global model is CESM and the regional model is WRF. The authors verified that the WRF model performs better than CESM in simulating historical extreme precipitation, and further used WRF model to project the future change in extreme precipitation after dynamic downscaling. The possible cause for the spatial pattern of the change are explored, in terms of changes in atmospheric circulation anomalies and oceanic conditions. I think the work is a good attempt to understand future change in extreme precipitation based on dynamic downscaling, and the work can be accepted if the authors well address the following specific issues.
Specific Issues
1) This study explains the precipitation change in terms of "water vapor transport". Based on the definition of water vapor transport, water vapor transport is stronger if wind is stronger since specific humidity varies little, but it does not necessarily converts into precipitation. A more accurate diagnostic framework for precipitation variability is water vapor budget analysis (Chou et al. 2009; Seager et al. 2010). Many previous works have simplified the water vapor budget equation and shown that vertical velocity at mid troposphere dominates precipitation variability and the uncertainty of future precipitation change (Huang et al. 2013; Huang and Xie 2015), rather than water vapor amount. Therefore, I suggest that the authors revise the texts and pay more attention to the change in vertical velocity (rather than water vapor) when explaining the mechanism of precipitation change.
Chou C, Neelin JD, Chen C-A, Tu J-Y (2009) Evaluating the “Rich-Get-Richer” Mechanism in Tropical Precipitation Change under Global Warming. J Climate 22 (8):1982-2005. doi:10.1175/2008jcli2471.1
Seager R, Naik N, Vecchi GA (2010) Thermodynamic and Dynamic Mechanisms for Large-Scale Changes in the Hydrological Cycle in Response to Global Warming. J Climate 23 (17):4651-4668. doi:10.1175/2010jcli3655.1
Huang P, Xie S-P, Hu K, Huang G, Huang R (2013) Patterns of the seasonal response of tropical rainfall to global warming. Nature Geosci 6 (5):357-361. doi:10.1038/ngeo1792
Huang P, Xie S-P (2015) Mechanisms of change in ENSO-induced tropical Pacific rainfall variability in a warming climate. Nature Geosci 8 (12):922-926. doi:10.1038/ngeo2571
2) The authors explained that anomalous anticyclone induced descent over the Hunan-Hubei provinces, but the anomalous anticyclone (Fig. 7a) is associated with descent anomaly on its southern flank and ascent anomaly on its northern flank (Fig. 7c). Although anticyclone in the boundary layer stimulates divergence and descent through Ekman pumping mechanism, anticyclone in the free troposphere has no direct relation to vertical motion. For example, ascent prevails within the South Asian high, and ascent also prevails on the western edge of the western Pacific subtropical high as seen in climatology. In the free troposphere, vertical motion (and its anomaly) is mostly generated by quasi-horizontal motion moving along sloping isentropic surfaces (e.g., Hoskins et al. 2003; Nie et al. 2020; Wu et al. 2020). In winter, isentropic surfaces incline northward with altitude over the YRB, and southerly (northerly) wind anomaly generates ascent (descent) anomaly, just as in most mid-latitude regions (He 2023). But in summer, there is a climatological warm center over the Tibetan Plateau and the isentropic surfaces incline eastward with altitude over YRB, therefore, westerly (easterly) wind anomaly stimulates ascent (descent) anomaly over YRB region in summer (He et al. 2023). As seen in Fig. 7a,c, the anomalous descent (ascent) is located on the southern (northern) flank of the anomalous anticyclone where anomalous easterly (westerly) wind prevails, so the anomalous descent (ascent) in Fig. 7c can be well explained by the easterly (westerly) wind anomaly on the southern (northern) side of the anomalous anticyclone via the isentropic gliding mechanism (He et al. 2023).
Reference
Hoskins B, Pedder M, Jones DW (2003) The omega equation and potential vorticity. Q J Roy Meteor Soc 129 (595):3277-3303. doi:10.1256/qj.02.135
Nie J, Dai P, Sobel AH (2020) Dry and moist dynamics shape regional patterns of extreme precipitation sensitivity. Proceedings of the National Academy of Sciences:201913584. doi:10.1073/pnas.1913584117
Wu G, Ma T, Liu Y, Jiang Z (2020) PV‐Q Perspective of Cyclogenesis and Vertical Velocity Development Downstream of the Tibetan Plateau. Journal of Geophysical Research: Atmospheres 125 (16):e2019JD030912. doi:10.1029/2019jd030912
He C, Zhou T, Zhang L, Chen X, Zhang W (2023) Extremely hot East Asia and flooding western South Asia in the summer of 2022 tied to reversed flow over Tibetan Plateau. Clim Dynam. doi:10.1007/s00382-023-06669-y
3) The authors stated that the western Pacific subtropical high (WPSH) has strengthened in recent decades (L111-116). However, this may be a spurious phenomenon arising from the systematic rising trend of global geopotential height (gph) which is almost proportional to temperature according to hypsometric equation (He et al. 2015, 2018). Indeed, previous studies have revealed a weakening trend of the WPSH in recent decades by directly examining atmospheric circulation (Huang et al. 2015; Wu and Wang 2015).
In the main text, the authors also stated that they are trying to explain the change in vertical motion in terms of the change in WPSH by examining the change in gph (L337-341), but the results shown in Fig. 7 only focus on East Asian continent, rather than western Pacific where WPSH is located. Since the change in local vertical velocity can be well explained and Fig. 7did not show the change in WPSH, I suggest the authors revise the corresponding text in L337-341 to avoid confusing the readers.
He C, Zhou T, Lin A, Wu B, Gu D, Li C, Zheng B (2015) Enhanced or Weakened Western North Pacific Subtropical High under Global Warming? Scientific Reports 5:16771. doi:10.1038/srep16771
Huang Y, Wang H, Fan K, Gao Y (2015) The western Pacific subtropical high after the 1970s: westward or eastward shift? Clim Dynam 44 (7-8):2035-2047. doi:10.1007/s00382-014-2194-5
Wu L, Wang C (2015) Has the Western Pacific Subtropical High Extended Westward since the Late 1970s? J Climate 28 (13):5406-5413. doi:10.1175/jcli-d-14-00618.1
4) The authors try to explain the change in atmospheric circulation in terms of SST anomaly and PDO. However, Fig. 8 does not show evident signal in the ocean, and extra-tropical SST anomalies are typically passive forced response to atmospheric circulation anomalies. Without solid evidence about the role of PDO, I suggest that Fig. 8 and most of the discussion about PDO are deleted. Some briefly discussion about possible future work on the atmosphere-ocean interaction is enough. The texts in the 3rd major conclusion in Section 5 and the abstract need to be revised, accordingly.
5) Downscaling based on regional models is to obtain added value compared with global models. What added value is obtained based on the WRF downscaled projection, in comparison with the projected change based on CESM? Some discussion is needed, and it would be helpful if additional figures is supplied to show a comparison of projected changes between regional model and global model.
The English expression is good.
Reviewer 3 Report
Liu et al. “Spatio-Temporal Analysis of Simulated Summer Extreme Precipitation Events under RCP4.5 Scenario in the Middle and Lower Reaches of the Yangtze River Basin.”
This manuscript aims to investigate the importance of downscaling method from 1 degree to 0.25 degree using the WRF model, and the data are reliable in simulating the summer extreme precipitation events over the middle and lower reaches of the Yangtze River Basin. A comparison between CESM1.0 27 data and WRF_025 data was discussed. The results show that WRF_025 data significantly improves the ability to simulate the numerical value and distribution of extreme summer precipitation in the middle and lower reaches of the Yangtze River. This study's results are interesting and valuable to further the discussion. Overall, the level of the scientific contribution of this work has met the journal with minor changes needed. I recommend that the paper be accepted with minor revisions.
Comments:
1. Line 178-179, Change the pm to UTC.
2. Table 1. Parameterization scheme. What is the reason for choosing the microphysics process WSM3 simple ice? There are multiple microphysics options available for high-resolution precipitation forecasts.
3. Line 240-244; “The typical spatial distribution ……… typical spatial structure of variable fields”. Check the English grammatical and rewrite the whole section.
4. Section-4 Discussion has only a few lines. Combining the discussion and conclusion in the same section will be good.
5. Conclusion section, write a few lines about the importance of high-resolution data assimilation compared to simply downscaling data from lower resolution to higher resolution, and with appropriate references.
The English language needs to improve.
